# OpenReview forum: "GEM: Geometric Erasure by Contrastive Velocity Matching in Rectified Flows"
_ICML.cc/2026/Conference — ICML 2026 spotlight_

### Official Review · Reviewer_Tbzt · 2026-03-11

**Soundness:** 3
**Presentation:** 3
**Significance:** 3
**Originality:** 3
**Overall Recommendation:** 5
**Confidence:** 3

**Summary:**

This paper proposes GEM (Geometric Erasure by Contrastive Velocity Matching), a concept-erasure method designed for rectified-flow text-to-image models such as FLUX and SD 3. The author shows that the trajectory-based objective used in ERASEFLOW can be approximated as a teacher-guided velocity matching loss. Building on this connection, GEM introduces a geometric contrastive objective that attracts model toward safe trajectories and repels it from unsafe ones. The method is evaluated on safety-related concepts (nudity, gore) and rights-protected concepts (celebrities and copyrighted characters). Experiments show improved erasure performance and faster training compared to prior approach.

**Compliance With Llm Reviewing Policy:**

Affirmed.

**Final Justification:**

Thank you to the authors for the rebuttal. My concerns have been addressed, so I have raised my rating.

**Key Questions For Authors:**

If the concerns are addressed, I would be willing to further raise the rating to 5 or 6.

**Limitations:**

Discussing the limitations or failure cases would further strengthen the work.

**Strengths And Weaknesses:**

### Strengths
1. Concept erasure method are mostly designed for U-Net diffusion models, while modern systems increasingly rely on rectified-flow transformers. Addressing gap is important.
2. The paper provides a clear derivation connecting trajectory-based erasure (GFlowNet perspective) and teacher-guided velocity matching, offering a unified view of these methos.
3. Across safety and rights-protection benchmarks, GEM reduce unsafe generation rates and often outperforms ERASEFLOW while being significantly faster.

### Weakness
1. Experiments focus mainly on safety and identity removal. Additional evaluations on other concept types (e.g., styles or object) would strengthen the claims of generality.
2. Incorporating experiments against attack methods like UD [1], RAB [2], etc would strengthen the work.
3. The paper compares against several representative baselines (ESD and UCE). However, it would strengthen the empirical evaluation if the authors could also include MACE as a baseline.
4. The authors could improve the paper's comprehensiveness by incorporating a discussion of related work, ANT [3].
5. Discussing the limitations or failure cases would further strengthen the work.

[1] To generate or not? safety-driven unlearned diffusion models are still easy to generate unsafe images... for now

[2] Ring-a-bell! how reliable are concept removal methods for diffusion models?

[3] Set You Straight: Auto-Steering Denoising Trajectories to Sidestep Unwanted Concepts

---

> ### Author Rebuttal · Authors · 2026-03-30
>
> We thank the reviewer for highlighting our clear derivation linking trajectory-based erasure and teacher-guided velocity matching, and for recognizing the strong empirical performance and efficiency of GEM across both safety and rights-protection benchmarks.
>
> ---
> # W1 - Additional Object Erasure Experiments
> To address this, **we added two additional object-erasure experiments on FLUX: erasing ✗Airplane while measuring retention on Ship, and erasing ✗Dog while measuring retention on Cat**.
> ||✗ Airplane ↓|Retention (Ship) ↑|✗ Dog ↓|Retention (Cat) ↑|Time (min) ↓|
> |:-|-:|-:|-:|-:|:-:|
> |FLUX|100|100|98.67|100|0|
> |ESD|7.00|40.67|54.67|62.33|34:41|
> |UCE|25.67|**98.67**|61.00|**99.67**|**0:14**|
> |MACE|21.33|97.33|57.67|97.67|5:48|
> |EraseFlow|6.33|61.67|0.00|88.00|18:47|
> |GEM (Ours)|**2.67**|77.00|**1.33**|99.33|4:02|
>
> **These results support the broader applicability of GEM. For ✗ Airplane, GEM achieves the strongest erasure while retaining substantially more than ESD and EraseFlow**. UCE and MACE preserve retention better, but at the cost of much weaker erasure, consistent with the trade-off discussed in our response to reviewer o2nU (W4). For ✗ Dog, GEM again achieves near-complete erasure (1.33) while preserving Cat retention at 99.33. We will include these additional object-level results in the revision.
>
> ---
> # W2 & W3 - Adversarial Robustness & Additional Baselines
> We agree that adversarial attack evaluations strengthen the paper and that MACE is a legitimate baseline. **Adversarial prompts generated with the RAB method were already included in the main submission, but to further address this point, we have now added two additional adversarial prompt benchmarks, MMA and Prompting4Debugging (P4D)**. All three settings are designed to elicit erased concepts from the edited model through adversarially constructed prompts rather than standard benchmark prompts. **We also expanded the baseline set by adapting both MACE and Concept Ablation (CA) (Kumari et al., 2023) to FLUX**.
> ||I2P ↓|T2I-RP ↓|RAB ↓|MMA ↓|P4D ↓|Basic ↓|FID ↓|min ↓|
> |-|:-:|:-:|:-:|:-:|:-:|:-:|-|-|
> |*FLUX*|*20.20*|*51.60*|*63.86*|*27.40*|*47.43*|*77*|*0.00*|*0*|
> |ESD|17.62|46.89|62.11|14.00|41.54|56|4.12|32:26|
> |CA|12.43|32.84|47.19|24.10|50.36|53|7.12|35:14|
> |UCE|18.69|49.29|55.44|16.90|38.60|73|2.47|**0:12**|
> |MACE|19.44|49.74|58.11|18.60|42.34|71|**2.14**|6:48|
> |EraseAnything|17.73|45.20|48.42|12.60|34.92|42|3.81|\-|
> |EraseFlow|9.77|36.66|42.46|6.70|17.28|42|8.32|15:58|
> |**GEM (Ours)**|**6.77**|**19.63**|**28.77**|**1.70**|**16.17**|**10**|8.20|3:27|
>
> The results remain consistent with our main conclusions. **GEM achieves the strongest overall performance on the adversarial evaluations, with the lowest unsafe rates on RAB, MMA, and P4D, while also outperforming the newly added MACE baseline on these settings**. We have additionally submitted a request for evaluation under the UnlearnDiffAtk protocol [1], and will include those results in the final version.
>
> ---
> # W4  - Comparison to ANT
> We appreciate the pointer as ANT is related and will be added to the discussion in the revision. ANT proposes a teacher-guided concept-erasure procedure that applies different guidance behaviors across the generation process, using ESD-style negative guidance to suppress the concept. Their method differs from GEM in several important respects. First, **ANT was developed in the context of earlier Stable Diffusion models trained with score matching and typically used with stochastic samplers, whereas GEM is designed for rectified-flow models** and their deterministic generation dynamics. In terms of methodology, **ANT stays within the standard teacher-guided setting where updates are applied at isolated timesteps sampled across trajectories, while GEM leverages consecutive timesteps from the same sampled trajectory** to provide trajectory-consistent supervision. **ANT also relies solely on target-derived (negative) guidance, whereas GEM combines this negative signal with an explicit safe-anchor pull** via the proposed geometric contrastive objective. Finally, **ANT uses a multi-term objective with several hyperparameters, while GEM uses a single contrastive formulation with one main trade-off parameter**, simplifying tuning.
>
> ---
> # W5 - Limitations
> We agree that a discussion of limitations is important and will expand it in the revision. An underexplored limitation in the concept-erasure literature is the bias induced by the erasure objective itself. In teacher-guided erasure, **a shared semantic bias across fine-tuning samples may let the model reduce loss by consistently encoding that bias rather than removing the intended concept**. Our method partly mitigates this through a geometric loss with both attractive and repulsive directions instead of a single target signal. Still, a more systematic analysis, including metrics sensitive to subtle semantic shifts not captured well by coarse measures such as FID, is needed to better understand these residual effects.

---

> > ### Author Rebuttal · Reviewer_Tbzt · 2026-04-04
> >
> > Thank you to the authors for the rebuttal. My concerns have been addressed, so I have raised my rating.

---

> > > ### Author Response · Authors · 2026-04-04
> > >
> > > We are thankful that our response resolved the concerns and we appreciate the reviewer’s support for acceptance.

---

### Official Review · Reviewer_234q · 2026-03-12

**Soundness:** 3
**Presentation:** 2
**Significance:** 3
**Originality:** 3
**Overall Recommendation:** 4
**Confidence:** 4

**Summary:**

The paper introduces GEM, an erasure framework specifically designed for Rectified Flow Transformers. Their proposed method utilizes a contrastive velocity matching objective that aligns the model’s predicted velocity with an anchor concept velocity to push it away from the target concept.

**Compliance With Llm Reviewing Policy:**

Affirmed.

**Key Questions For Authors:**

1. How sensitive is the model to $\eta$ in Equation 14. If you have already analysed this, and I have missed it, please let me know.
2. Is there a latency overhead compared to other trajectory-based methods like EraseFlow?

**Limitations:**

I believe that I have highlighted some limitations in the Weaknesses section. However, I urge the authors to highlight limitations of their method in the manuscript.

**Strengths And Weaknesses:**

**Strengths:**
1. The paper is well written.
2. The proposed method is technically sound.
3. The paper demonstrates strong erasure performance that outperforms previous baselines.


**Weaknesses:**
1. Lack of adversarial attacks - A major experiment in recent Concept Erasure works is evaluation against white/block-box adversarial attacks like [1,2, 3]. This is because many concept erasure methods are not robust to white-box attacks and may regenerate the target concept with adversarial prompt embeddings [3]. Can you add some results evaluating your model against the attacks mentioned?
[1] To Generate or Not? Safety-Driven Unlearned Diffusion Models Are Still Easy To Generate Unsafe Images ... For Now, Zhang et al
[2] Prompting4debugging: Red-teaming text-to-image diffusion models by finding problematic prompts, Zhang et al
[3] Xiaosen Wang, Nan Xu, and Qiang Xu. M, Yang et al

2. Multi-concept Erasure - Is it possible to erase multiple concepts from the model using your method? For example, UCE and other related works show multi-concept erasure experiments where they erase several artist styles from the model simultaneously. Can you provide some small-scale results for this scenario?

3. Does erasure of a particular concept unintentionally lead to erasure of a very similar concept? For example, Erasing 'dog' may affect the generation of cats. It has been seen that erasing Van Gogh style often affects Claude Monet's style. Does your method prevent this?

---

> ### Author Rebuttal · Authors · 2026-03-30
>
> We thank the reviewer for recognizing the method's technical soundness, the clarity of the presentation, and its strong empirical erasure performance relative to prior baselines.
>
> ---
> # W1 - Lack of Adversarial Evaluation
> We agree that robustness against adversarial prompting is an important part of evaluating concept erasure, and note that **RAB in the main paper already evaluates adversarial prompts optimized to bypass erasure. To address this further, we extended the FLUX safety evaluation (Table 2 in the main paper) with two attack settings: MMA [3] and P4D [2]**.
> ||I2P ↓|T2I-RP ↓|RAB ↓|MMA ↓|P4D ↓|Basic ↓|FID ↓|Time (min) ↓|
> |-|:-:|:-:|:-:|:-:|:-:|:-:|-|-|
> |*FLUX*|20.20|51.60|63.86|27.40|47.43|77|0.00|0|
> |ESD|17.62|46.89|62.11|14.00|41.54|56|4.12|32:26|
> |UCE|18.69|49.29|55.44|16.90|38.60|73|2.47|**0:12**|
> |EA|17.73|45.20|48.42|12.60|34.92|42|3.81|\-|
> |EraseFlow|9.77|36.66|42.46|6.70|17.28|42|8.32|15:58|
> |**GEM (Ours)**|**6.77**|**19.63**|**28.77**|**1.70**|**16.17**|**10**|8.20|3:27|
>
> The results remain consistent with the main paper: GEM achieves the lowest unsafe rates among all compared methods on both attacks, with 1.70 on MMA and 16.17 on P4D, improving over EraseFlow (6.70, 17.28) and substantially over teacher-guided baselines such as ESD, UCE, and EraseAnything. We have additionally submitted a request for evaluation under the UnlearnDiffAtk protocol [1], and will include those results in the final version.
>
> ---
> # W2 - Multi-concept Erasure
> **Yes, GEM can be extended to multi-concept erasure by sampling`(target, anchor)` pairs from a concept set during training**, but it requires additional care, since joint editing can lead to over-erasure and broader distribution shift. To test this, we conducted a small-scale experiment erasing a subset of artistic styles from Lu et al. (2024). We added regularization on the unconditional prediction, at the cost of a small runtime increase. We evaluate retention with CLIP on a held-out set of 5 artists and compare the performance against UCE.
> |#Artists|Method|Erasure (Avg.) ↓|Retention (Avg.) ↑|
> |:-:|:-|:-:|:-:|
> |1|UCE|16.50|25.93|
> |1|GEM|16.43|25.67|
> |5|UCE|19.70|26.62|
> |5|GEM|20.18|25.27|
>
> **GEM remains competitive with UCE in this multi-concept setting**. As the number of erased concepts increases, both methods show some degradation, with UCE holding a slight advantage in this experiment. Scaling to larger concept sets will likely require stronger regularization, multi-adapter fusion strategies as in Lu et al. (2024), or explicit preservation concepts.
>
>
> ---
> # W3 - Unintended Collateral Erasure
> **We do evaluate this type of collateral damage in the main paper through the *Retention* metric**. For each erased concept, we include related concepts from the same category that should remain unaffected. For example, when erasing Albert Einstein we test retention on other public figures, and when erasing Son Goku we test retention on another anime character, Naruto. In these rights-protection settings, GEM achieves the strongest or near-strongest retention among the compared methods, depending on the backbone.
> To address the reviewer’s dog/cat example more directly, **we additionally ran an object-level experiment where we erase Dog and measure retention on Cat**.
> ||Erasure (Dog) ↓|Retention (Cat) ↑|Time (min) ↓|
> |:-|-:|-:|:-:|
> |FLUX|98.67|100.00|0|
> |ESD|54.67|62.33|34:41|
> |UCE|61.00|**99.67**|**0:14**|
> |EraseFlow|0.00|88.00|18:47|
> |GEM (Ours)|**1.33**|99.33|4:02|
>
> **The results show that GEM achieves near-complete erasure (1.33) while preserving cat at 99.33**. This is substantially better than ESD and EraseFlow and comparable to the strongest retention-preserving baseline, UCE (99.67) while maintaining much stronger erasure.
>
> ---
> # Q1 - $\eta$ - Analysis
> Yes, **we analyze sensitivity to $\eta$ in the supplementary material, specifically Sec. D.1 (“Ablation of $\eta$”)**, under a fixed compute budget of 250 steps with $t_{\mathrm{stop}} = 5$. In our formulation, $\eta$ is the main control for the repulsion strength in Eq. 14, and the ablation over $\eta \in \{0.00, 0.30, 0.50, 0.75, 0.80, 0.90, 1.00\}$ shows a trade-off: increasing $\eta$ generally strengthens erasure, especially for the more challenging nudity benchmark, but large values can also increase utility shift as reflected in FID (e.g., FID of 16.02 at $\eta=1.00$). We will make this more explicit in the revised main paper and add a short summary of the key trend to improve visibility.
>
> ---
> # Q2 - Latency Comparison
> **We report wall-clock erasure times in the evaluation tables** (last column), and GEM does not introduce a latency overhead relative to trajectory-based methods such as EraseFlow. On the contrary, GEM is consistently faster in our experiments. For example, on copyrighted character erasure with FLUX, EraseFlow requires 7:48 minutes, while GEM takes 1:23 minutes. More broadly, across our evaluation settings, **GEM reduces erasure time by about 70.6% on average relative to EraseFlow**.

---

> > ### Author Rebuttal · Reviewer_234q · 2026-04-03
> >
> > The rebuttal addresses most of my concerns. I will maintain my rating - Weak accept

---

> > > ### Author Response · Authors · 2026-04-04
> > >
> > > We thank the reviewer for the positive acknowledgment and are glad that the rebuttal helped to clarify the concerns. If the reviewer deems appropriate, we would appreciate reconsideration of the score to reflect this updated perspective.

---

### Official Review · Reviewer_o2nU · 2026-03-12

**Soundness:** 3
**Presentation:** 3
**Significance:** 2
**Originality:** 2
**Overall Recommendation:** 4
**Confidence:** 4

**Summary:**

This paper studies the problem of concept removal for Rectified Flows instead of DiT-based models. Based on ERASEFLOW, this paper proposes a Geometric Erasure Method (GEM) by establishing a bridge between trajectory-based objectives and teacher-guided velocity matching, and deriving a geometric contrastive loss for using selective timestep scheduling. The method demonstrates strong empirical suppression of explicit content on the FLUX model.

**Compliance With Llm Reviewing Policy:**

Affirmed.

**Final Justification:**

After considering the rebuttal and other reviewers' comments, I am revising my recommendation from weak reject to weak accept. The paper studies an important and timely problem, namely concept erasure for rectified flow models, and proposes a simple geometric contrastive objective that is intuitive and practically useful. The empirical results are promising, especially for FLUX, and the paper has clear practical relevance. My main concerns were about the rigor of the theoretical claim, the degree of novelty relative to prior work such as EraseFlow, and the limited evidence for generality. The rebuttal addressed these concerns in a constructive way. In particular, the authors clarified that the theoretical bridge should be viewed as an approximate but grounded connection rather than a fully rigorous equivalence, and they added new experiments on Qwen-Image and quantization robustness that strengthen the paper empirically.

**Key Questions For Authors:**

Please refer to the weaknesses part above.

**Limitations:**

Partially. It would be great if some discussions about concept restoration are provided, e.g., quantization.

**Strengths And Weaknesses:**

**Strengths**
- S1. The paper studies that problem of concept erasure for the latest generation of Rectified Flow models (e.g., FLUX), which haven't been received as much attention as standard diffusion models.
- S2. The proposed geometric contrastive loss (pulling the student toward a safe anchor and pushing it away from an unsafe target) is easy to implement and reasonable.

**Weaknesses**
- W1.The key claims of this paper is that it constructs a theoretical bridge between the trajectory-based objectives and teacher-guided velocity matching. Nevertheless, the derivation mainly relies on the empirical leaps. For example, in Step 2, it is stated that a performance drop is observed when $\log Z_\phi > \log \beta$, so the offset is absorbed and replace the squared-residual minimization with a maximum-likelihood approximation. That's a heuristic patch to make the math work instead of a rigorous reduction.

- W2. The proposed method heavily relies on previous work, i.e., ERASEFLOW. The main difference is Step 5 but it borrows the idea of the selective timestep scheduling from Lu et al. (2024). It feels less like a structurally novel erasure mechanism and more like an empirically-tuned contrastive loss paired with existing scheduling tricks.

- W3. This paper claims GEM is a general approach for Rectified Flow models, but it mainly conducts experiments with FLUX. The evaluation on SD3 is few with a small-scale dataset.

- W4. The baseline UCE method has noticeably better utility preservation and faster execution times. While GEM has a better unsafe Rate, it is more like a tradeoff since the unsafe Rate is not close to 0.

- W5. Recent studies show that the concept erase can be recovered by quantizing the models. It is suggested to conduct experiments on some defending techniques for concept removal.

---

> ### Author Rebuttal · Authors · 2026-03-30
>
> We thank the reviewer for recognizing the importance of extending concept erasure to modern rectified-flow models and for highlighting the simplicity and soundness of the proposed geometric loss.
> # W1 - Non-rigorous Reduction
> **The paper does not claim an end-to-end rigorous equivalence between trajectory-balance (TB) training and teacher-guided velocity matching**. The stated claim is explicitly weaker: the bridge is presented as a sequence of theoretical and empirical reductions, yielding a **theoretically grounded approximation** that is useful for designing practical objectives.
> Concretely, the mentioned “empirical leap” is grounded in observations reported by the EraseFlow authors. They identify the gap $(\log\beta-\log Z_\phi)$ as the decisive factor controlling when probability flow is sufficiently redirected.
> The intent of Step 2 is therefore not to “patch the math”, but to formalize what the objective is effectively doing in the empirically relevant regime used by EraseFlow.
> # W2 - Heavy Reliance on Prior Work
> GEM is informed by EraseFlow, but it is not a minor variation. **The prior is derived from a TB objective motivated by probability-mass transport, whereas GEM is a teacher-guided method based on velocity matching with a geometric contrastive signal**.
>
> While Lu et al. (2024) were among the first to explore timestep scheduling for concept erasure, GEM’s strategy is fundamentally different. Their method samples *individual timesteps*, whereas GEM samples *partial trajectories* to preserve conditional dependence between consecutive steps. Table 9 shows that this is not cosmetic: it improves erasure, but overly long segments can reduce fidelity, motivating GEM’s operating point between isolated-timestep and full-trajectory extremes.
>
> We agree that Step 5 captures the main methodological differences. The original teacher-guided erasure method ESD draws on the negative-guidance principle to move generations away from the harmful data manifold. In contrast, EraseFlow-style trajectory erasure redirects probability mass toward a safe trajectory and therefore relies on a positive anchor rollout by design. Since both approaches rely on a single signal, they admit shortcuts such as drift toward a constant semantic bias. GEM instead combines attraction with repulsion to keep trajectory-level erasure stable and effective.
> Importantly, this is not simply a matter of inserting one feature from one framework into another. **As our ablations in Supplementary D suggest, naively introducing a negative signal into the trajectory-based objective led to a substantial drop in model utility (Table 7: FID 16.02 vs. 3.51 without repulsion)**. This highlights the challenge of integrating the benefits of one framework into another without losing the strengths of the base paradigm. We will revise the paper to make this distinction clearer.
> # W3 - Limited Generality Evidence
> **To strengthen the generality claim, we add experiments on a third rectified-flow model, Qwen-Image**.
> ||I2P ↓|T2I-RP ↓|RAB ↓|Basic ↓|FID ↓|
> |-|:-:|:-:|:-:|:-:|:-:|
> |*Qwen-Image*|21.48|63.41|94.04|100|0.00|
> |ESD|17.51|47.98|53.33|**2**|12.63|
> |CA|19.66|54.17|55.44|9|8.86|
> |EraseFlow|17.08|40.05|57.89|18|8.18|
> |**GEM**|**9.88**|**20.16**|**26.32**|8|**7.91**|
>
> GEM achieves the best overall performance on 3 of the 4 benchmarks. While ESD outperforms GEM on Basic, it comes at the cost of reduced utility (FID 12.63 vs. 7.91).
> # W4 - Safety-Utility Tradeoff
> The safety-utility trade-off is inherent to concept erasure. Small parameter updates keep the edited model closer to the original distribution, lowering FID but also preserving more unsafe behavior. Stronger interventions can reduce unsafe generations, but may degrade image quality enough that detector scores become unreliable.
> **To expose the Pareto frontier, we ran GEM with  $\eta = 0.25$, for FLUX nudity erasure. This yields utility close to UCE, while still achieving stronger erasure on 3 of the 4 benchmarks**:
> ||I2P ↓|T2I-RP ↓|RAB ↓|Basic ↓|FID ↓|
> |-|:-:|:-:|:-:|:-:|:-:|
> |*FLUX*|20.20|51.60|63.86|77|0.00|
> |UCE|18.69|49:29|55.44|73|2.47|
> |**GEM** (η=0.25)|15.79|38.01|58.25|56|2.79|
> # W5 - Vulnerability to Quantization Recovery
> **To address this, we quantized the edited models to 4-bit precision** and measured the change in unsafe rate:
> ||I2P ↓|T2I-RP ↓|RAB ↓|Basic ↓|
> |-|:-:|:-:|:-:|:-:|
> |*FLUX*|+1.39|+1.33|+10.18|+6|
> |ESD|-0.97|+0.36|-0.71|0|
> |UCE|-1.07|+1.60|+2.10|+6|
> |EA|+0.85|+0.45|-0.35|0|
> |EraseFlow|-2.04|-7.89|-3.51|+1|
> |**GEM**|+1.29|-0.26|-7.72|-8|
>
> Quantization does not uniformly recover erased concepts. For some baselines, unsafe rates indeed increase again after quantization, indicating re-emergence of harmful behavior. **In contrast, the two methods that incorporate trajectory-based ingredients, EraseFlow and GEM, remain comparatively robust, and in several cases, quantization even further reduces unsafe rates**. We recognize this phenomenon as an important direction for future work.

---

> > ### Author Rebuttal · Reviewer_o2nU · 2026-04-02
> >
> > The rebuttal addresses several of my concerns. I appreciate the additional experiments on Qwen-Image, the quantization analysis, and the clarification of the safety utility tradeoff. These additions strengthen the empirical support of the paper and improve the overall presentation. My concerns are therefore partially resolved, though not fully. In particular, the clarification on the theoretical side is helpful, but it also suggests that the claimed bridge is better viewed as an approximate and practically motivated connection rather than a fully rigorous derivation.
> >
> > Overall, the rebuttal improved my assessment, and I believe the paper is stronger after these clarifications.

---

> > > ### Author Response · Authors · 2026-04-04
> > >
> > > We sincerely thank the reviewer for the careful reassessment and greatly appreciate the support for acceptance.

---

### Official Review · Reviewer_UjbX · 2026-03-12

**Soundness:** 2
**Presentation:** 2
**Significance:** 2
**Originality:** 3
**Overall Recommendation:** 4
**Confidence:** 4

**Summary:**

This paper proposes GEM, a concept-erasure method for rectified-flow T2I models (mainly FLUX, with a small SD3 validation). The method starts from ERASEFLOW’s trajectory objective, argues that in deterministic rectified flows it can be approximated by a teacher-guided velocity-matching loss, and then introduces a contrastive geometric loss that pulls the edited model toward a safe anchor velocity while repelling it from the unsafe target velocity.

**Compliance With Llm Reviewing Policy:**

Affirmed.

**Final Justification:**

The rebuttal addresses several of my concerns. I appreciate the authors’ clarifications and will maintain my rating as "Weak accept."

**Key Questions For Authors:**

Please see the weakness

**Limitations:**

yes

**Strengths And Weaknesses:**

## Strengths
- The steup is interesting, and the proposed loss is simple and practically efficient.
- Empirical results on FLUX are strong, especially for nudity/gore safety benchmarks.
- The paper includes both safety and rights-protection settings.

## Weaknesses
- The main practical gain may come from the final contrastive loss, not the claimed unification. The paper spends substantial effort on the bridge from ERASEFLOW to teacher-guided matching, but the actual performance gains seem more attributable to the proposed hinge-like contrastive objective and selective timestep training.

- Evaluation is narrow in some important ways. Most strong results are on FLUX and use detector-based unsafe rates. Rights-protection evaluation relies on Gemini as a black-box classifier, which is convenient but noisy and not especially rigorous.

- The paper compares against available methods, but several stronger robust erasure methods from the diffusion literature are absent, even if adaptation is difficult. This weakens the “state of the art” claim.

---

> ### Author Rebuttal · Authors · 2026-03-30
>
> We thank the reviewer for the feedback and for recognizing the practicality of the proposed loss, the strong FLUX results, and the breadth of evaluation settings.
>
> ---
> # W1 - Unclear Source of Gains
> We agree that **the strongest practical gains come from the contrastive objective together with selective training on early consecutive timesteps**. The intention is not to dispute this, but to clarify the role of the unification: it shows that ideas from trajectory-based erasure and teacher-guided matching are not isolated and limited to the corresponding framework. We show that useful design elements from one paradigm can, in principle, be transferred to the other. At the same time, this transfer is not automatic. **Because the two paradigms operate under different training dynamics, a naive port of one mechanism into the other can easily lead to instability, weaker erasure, or degraded fidelity (cf. Supplementary D1 and D3)**. This is precisely why the unification matters: it is useful not because it alone guarantees better performance, but because it reveals which ingredients are theoretically compatible and how they can be combined in a principled way.
>
> This is also how we view the two components highlighted by the reviewer. Our partial-trajectory training is motivated by the trajectory balance (TB) perspective, but instead of using all samples from a full rollout as the TB equation would suggest, **GEM leverages shorter consecutive segments to remain stable and effective in the teacher-guided setting**. Importantly, to the best of our knowledge, **no other teacher-guided method leverages multiple guidance signals from the same trajectory.**
>
> Similarly, the contrastive objective follows from viewing the two frameworks as lying on a shared spectrum rather than as disjoint approaches. The classic teacher-guided erasure method ESD, draws on the negative-guidance principle to move generations away from the harmful data manifold. In contrast, EraseFlow-style trajectory erasure redirects probability mass toward a safe anchor trajectory and is therefore based on an anchor rollout by design. Both approaches rely on a single signal, making them prone to shortcuts such as drift toward a constant semantic bias. In our work, the hyperparameter $\eta= 0$ corresponds to pure anchor signal as in the EraseFlow-style trajectory erasure, while larger $\eta$ moves the balance toward the teacher-guided ESD style objective. **Table 9 (Supplementary) jointly ablates both spectra and highlights that either extreme can degrade model fidelity or weaken erasure strength**.
> We will revise the paper to make this distinction clearer.
>
> ---
> # W2 - Black-box Judge Limitation
> **To reduce reliance on a single black-box VLM, we conducted a small-scale rights-protection evaluation using an ensemble of three open-source vision-language models**: Qwen3-VL-8B-Instruct, Gemma3-12B-it, and Pixtral-12B-2409, using the average prediction across the three models as the evaluation signal. We applied this evaluation to copyrighted character erasure for Stitch on both SD3 and FLUX.
>
> |SD3|Original|ESD|EraseFlow|GEM (Ours)|
> |-|:-:|:-:|:-:|:-:|
> |✗ Stitch (↓)|*100.00*|16.00|**0.67**|1.67|
> |**Retention** (↑)|*99.67*|79.33|12.00|**94.00**|
>
> |FLUX|Original|ESD|CA|UCE|EraseFlow|GEM (Ours)|
> |-|:-:|:-:|:-:|:-:|:-:|:-:|
> |✗ Stitch (↓)|*99.00*|**0.00**|58.00|**0.00**|12.00|**0.00**|
> |**Retention** (↑)|*99.00*|96.56|**98.67**|**98.67**|96.22|97.44|
>
> The results are consistent with our main conclusions. **On SD3, EraseFlow and GEM both achieve strong erasure**, but GEM preserves retention much better (94.00 vs. 12.00). **On FLUX, GEM reaches complete erasure of the target** together with ESD and UCE, while maintaining strong retention (97.44).
>
> We provide additional results on Qwen-Image in our response to Reviewer **o2nU** (W3).
>
> ---
> # W3 - Missing Baselines
> To address this concern, we expanded the experiments by adapting **two additional methods to FLUX: MACE** (Lu et al., 2024) **and Concept Ablation (CA)** (Kumari et al., 2023). We evaluated both methods on the FLUX nudity-erasure setting. MACE performs similarly to UCE, consistent with the fact that it also relies on a closed-form style update. **CA offers a different trade-off, reaching slightly better utility comparable to GEM in terms of FID, but with clearly weaker erasure performance across all four safety benchmarks**.
>
> ||I2P (↓)|T2I-RP (↓)|RAB (↓)|Basic (↓)|FID (↓)|min (↓)|
> |:-|:-:|:-:|:-:|:-:|:-:|:-:|
> |*FLUX*|*20.20*|*51.60*|*63.86*|*77*|*0.00*|*0*|
> |ESD|17.62|46.89|62.11|56|4.12|32:26|
> |CA|12.43|32.84|47.19|53|8.12|35:14|
> |UCE|18.69|49.29|55.44|73|2.47|**0:12**|
> |MACE|19.44|49.74|58.11|71|**2.14**|6:48|
> |EraseAnything|17.73|45.20|48.42|42|3.81|\-|
> |EraseFlow|9.77|36.66|42.46|42|8.32|15:58|
> |**GEM (Ours)**|**6.77**|**19.63**|**28.77**|**10**|8.20|3:27|

---

> > ### Author Rebuttal · Reviewer_UjbX · 2026-04-02
> >
> > The rebuttal addresses several of my concerns. I appreciate the authors’ clarifications and will maintain my rating as "Weak accept."

---

> > > ### Author Response · Authors · 2026-04-03
> > >
> > > We thank the reviewer for their acknowledgment. We are glad the rebuttal addressed your concerns. Given that the issues appear to be fully resolved, we would kindly ask whether you might consider revisiting the score to reflect this updated view.

---

### Decision · Program_Chairs · 2026-04-30

**Decision:**

Accept (spotlight)

**Comment:**

This paper is about concept erasure framework Rectified Flow models (e.g., FLUX and SD3). It combines trajectory-based erasure and teacher-guided velocity matching with a geometric contrastive loss. Experiments on nudity/gore removal and rights protection demonstrate strong erasure performance across multiple benchmarks.

All four reviewers are supportive of acceptance. Initial concerns included robustness to adversarial attacks, missing baselines, generalization to multi-concept erasure, clarity on the source of performance gains, and reliance on prior work. The rebuttal successfully addressed most of these issues. Remaining concerns include limited evaluation on SD3, questions about theoretical rigor (with the proposed connection viewed as approximate rather than fully rigorous), and the safety–utility trade-off, where GEM yields worse FID compared to UCE.

Overall, the area chair recommends acceptance. The work is timely in addressing concept erasure for rectified flow models, and the proposed geometric contrastive velocity matching loss is simple, well-motivated, and empirically effective.